# Length Generalization via Auxiliary Tasks

**Pranjal Awasthi**
Google
pranjalawasthi@google.com

**Anupam Gupta**
NYU
anupam.g@nyu.edu

**Ravi Kumar**
Google
ravi.k53@gmail.com

## Abstract

*Length generalization*, the ability of sequence models to generalize to sequences longer than those encountered during training, remains a key challenge for transformers, especially in tasks requiring algorithmic reasoning. Existing theoretical understanding of length generalization is limited, often providing only asymptotic results or focusing on specific problem classes or architectural variants, while empirical approaches frequently rely on ad hoc and often fragile techniques.

In this work we introduce a novel framework for analyzing and proving length generalization bounds under specified, verifiable assumptions. A key outcome of the theory is the identification of a natural set of *auxiliary* tasks, intricately related to the primary task structure, such that strong performance on these auxiliary tasks, alongside the primary task, provably guarantees length generalization within the framework. This motivates a multi-task training procedure that explicitly optimizes performance on both the primary and the identified auxiliary tasks.

Empirical evaluations on a variety of synthetic benchmarks known to be challenging for length generalization, including sequence sorting, and reversal, demonstrate that our proposed method yields significant improvements in generalization to substantially longer sequences.

## 1 Introduction

Transformer architectures (Devlin et al., 2019; Vaswani et al., 2017) have become the cornerstone of modern sequence modeling, achieving state-of-the-art performance across a vast range of applications. However, a persistent and critical limitation is their often poor performance on *length generalization*—the ability to process sequences at test time that are significantly longer than those seen during training (Anil et al., 2022; Kazemnejad et al., 2023; Press et al., 2022; Tay et al., 2021). This deficiency is particularly evident in tasks demanding algorithmic reasoning or extrapolation, such as multi-digit arithmetic, symbolic manipulation, or processing long documents (Saxton et al., 2019; Zhou et al., 2024b). The failure to robustly generalize beyond the training length distribution not only limits practical applications, but also casts doubt on whether these models are truly learning underlying algorithms, or merely performing sophisticated pattern matching within a constrained length range. Despite considerable research effort, achieving reliable length generalization remains a fundamental open problem (Nye et al., 2021; Saxton et al., 2019; Zhao et al., 2024).

Efforts to address length generalization have yielded mixed results. Empirically, various techniques have been proposed, often involving modifications to positional encodings (PEs) (Anil et al., 2022; Saxton et al., 2019) or attention mechanisms. More recent works have found highly specialized PEs like FIRE (Li et al., 2024) or position coupling (Cho et al., 2024; McLeish et al., 2024) essential for tasks like addition. However, it has remained difficult to draw clear lessons from these works about when and why they work; moreover, the resulting length generalization can be fragile, highly sensitive to factors like data formatting (e.g., reversed sequences, index hints for addition), scratchpad formats, random weight initialization, and training data order. This fragility raises the question of

39th Conference on Neural Information Processing Systems (NeurIPS 2025).

whether these empirical fixes are merely learning task-specific shortcuts rather than inducing robust and generalizable algorithmic capabilities.

There have been several theoretical investigations into length generalization and how to boost performance (e.g., (Ahuja and Mansouri, 2024; Huang et al., 2025; Sabbaghi et al., 2024; Zhou et al., 2024a)), but our understanding remains limited. The most relevant prior works suggest that sparsity is a critical factor for length generalization (e.g., (Golowich et al., 2025; Huang et al., 2025; Sabbaghi et al., 2024)); we compare our work to these in Section 1.1.

**Our results.** We propose a novel framework for studying length generalization in transformers.

1. We present a general theoretical framework to study length generalization. Rather than specializing to particular tasks or training procedures, we aim for broad applicability and focus on the class of *sparse* tasks: loosely speaking, these are tasks where each output token depends on the presence of a small number of preceding tokens. (We formally define sparsity in Section 2.1.)

2. We show that robust performance at small training lengths implies generalization for such sparse tasks. Formally, our theoretical result considers input distributions $\{\mathcal{D}_k\}_{k \in \mathbb{Z}_+}$, where $\mathcal{D}_k$ is a distribution over length-$k$ input sequences. We show that if the transformer model is "robustly" correct (with error at most $\varepsilon$ and large margin) on sequences of length $n$, then it has error $\varepsilon \cdot N/n$ on sequences of length $N \gg n$—i.e., the error increases only linearly in the sequence length. Here, being robustly correct at length $n$ means that the error of the model is less than $\varepsilon$, not only on sequences drawn from $\mathcal{D}_n$, but also on suitable $n$-length random restrictions of sequences drawn from $\mathcal{D}_N$. (We present the notions of $(N, n)$-perturbations and robust correctness in Section 2.1.)

3. Our framework is not just descriptive, it also prescribes a *new* training procedure. Indeed, the above results identify robust correctness as a desirable property: one way to ensure it at length $n$ is to actually train the model not only on the "native" distribution $\mathcal{D}_n$ but also on random $n$-length restrictions of $\mathcal{D}_N$—we call this *robust training*. It is well-known that training on just $\mathcal{D}_n$ leads to poor length generalization, so our hope is that training on these random restrictions will lead to a robustly correct model that can length-generalize; this is supported by our formal results.

Moreover, many common sparse tasks generate the next token by applying one of a small set of functions to (a small subset of) the preceding tokens. For instance, the sorting task requires repeatedly finding the successor; likewise, the increment task repeatedly computes sums of two or three digits. We treat these functions as *auxiliary* tasks and train the model jointly on the original and auxiliary tasks, drawing data from both $\mathcal{D}_n$ and random $n$-length restrictions of $\mathcal{D}_N$.

4. Finally, we empirically validate our approach by conducting experiments on several tasks, including commonly studied synthetic tasks such as sorting and reversal. We also introduce new tasks, including the SLiM task, which is aimed to capture a toy language modeling scenario. Our results show that performing robust training at length $n$ leads to good performance at length $N \gg n$.

The structure of the paper is as follows: after reviewing related work in Section 1.1, we present the theoretical framework in Section 2. Our length generalization result is given in in Section 3, showing how robust correctness implies length generalization. Section 4 contains our robust training process inspired by the theory, and presents the results of our experiments. Finally, Section 5 contains concluding remarks and future directions.

## 1.1 Related Work

**Empirical Work.** Empirical studies have consistently shown that transformers exhibit mixed success in length generalization (Duan et al., 2023; Jelassi et al., 2023). While proficient at many sequence tasks, they often fail dramatically when tested on sequences longer than those seen during training, particularly on algorithmic tasks like multi-digit addition (Cho et al., 2024), sorting (Lee et al., 2024), sequence reversal, parity checking (Zhou et al., 2024b), and ListOps (Nangia and Bowman, 2018). These tasks serve as crucial benchmarks because their underlying algorithms possess inherent recursive structures that models should ideally learn to apply, irrespective of length.

There is much work on the role of *Positional Encoding (PE)*. Relative PEs were initially favored over absolute ones (Wang et al., 2024a), but subsequent works gave conflicting evidence. Some found that removing explicit PEs entirely (NoPE) could yield better length generalization on certain downstream tasks (Wang et al., 2024a), whereas others showed that highly task-specific PE schemes,

such as FIRE (Li et al., 2024) or position coupling (assigning identical position IDs to structurally related tokens) (Cho et al., 2024; McLeish et al., 2024), helped achieve length generalization on tasks like addition. Our work only uses NoPE, and it remains a fascinating question to see if it can be combined with more sophisticated PEs.

Data formatting and representation also play a critical role. Techniques like reversing the sequence order for addition (Lee et al., 2024; Li et al., 2024), giving explicit index hints within input/output sequences, or using intermediate scratchpads or Chain-of-Thought reasoning steps (Merrill and Sabharwal, 2024) can dramatically improve length generalization on specific tasks. Unfortunately, scratchpads are not a panacea—e.g., for the problems we consider, the baseline models (with scratchpads) do not length-generalize to the extent that our proposed models (also using scratchpads) do. Index hints and position coupling offer considerable power, but require the input to be fed with hints during *both* training *and* inference: our goal here is to investigate approaches where the input does not have to be augmented with hints at inference time. (It would be interesting to study if these ideas can be combined.) Other empirical approaches include modifications to the attention mechanism itself (Fan et al., 2025) or develop specialized training techniques (Lee et al., 2025).

Another potential approach to improving length generalization is to use curriculum learning (e.g., (Pouransari et al., 2024)) and train on progressively more complex instances that are also longer. Our work is once again orthogonal, and in principle, can be combined with curriculum learning; we leave this as an interesting future research direction.

**Theoretical Work.** Theoretical understanding of length generalization is an active area of research, with several perspectives emerging. One line of work aims to characterize the class of functions on which transformers length-generalize. The RASP-L Conjecture (Zhou et al., 2024a) posits that transformers length-generalize precisely on tasks solvable by short programs in RASP (a language that models transformer computation (Weiss et al., 2021)); despite recent work (Huang et al., 2025; Yang and Chiang, 2024), this conjecture remains open. Some other works have studied simpler models like linearized attention (Sabbaghi et al., 2024) or average pairwise attention (Ahuja and Mansouri, 2024), and show asymptotic length generalization for these classes. Huang et al. (2025) define the limit transformer and show length generalization asymptotically. Li et al. (2025) consider the "vanishing variance" phenomenon where attention output variance decreases with length under certain assumptions. Many other works focus on showing expressivity of transformers, and do not consider issues of learnability. Our work focuses on giving general conditions under which the model exhibits length generalization, and gives quantitative bounds for finite lengths.

The work of Wang et al. (2024b) proves that performing gradient descent on transformers successfully learns sparse token selection (i.e., the task of choosing a sparse set of $k$ out of $n$ tokens and taking their average); moreover it exhibits good length generalization, where the error scales as the square of the length increase. Although it relies on a specific training process, this result is very much in the same spirit as ours. A concurrent work focusing on sparsity is that of Golowich et al. (2025), who also propose a theoretical framework to study generalization via sparsity; they use a notion of *predictive position coupling* to achieve length generalization. This idea is orthogonal to ours, and it would be interesting to see if they can be combined. Finally, formal language theory offers a different perspective by investigating the classes of formal languages that transformers can recognize or generate (Strobl et al., 2024).

**Previous Version.** This paper substantially extends the preprint by Awasthi and Gupta (2023), which investigated task hinting as a mechanism for length generalization. Our theoretical analysis yields a new robust training framework based on auxiliary tasks—a component absent in the prior work. Furthermore, this new approach leads to improved empirical performance across a range of tasks. For instance, the increment task length-generalized (i.e., achieved $\geq 90\%$ test accuracy) by at most a factor of 1.2 in the previous work, whereas it does so by at least a factor of 1.5 in ours.

## 2  A Framework for Length Generalization

In this section we introduce a formal framework to analyze the conditions under which transformer models length-generalize on sequence-to-sequence tasks; we then prove the generalization bounds in Section 3.

Let $\Sigma$ and $\Gamma$ be finite input and output alphabets respectively. A *sequence-to-sequence* task is defined by a target function $f : \Sigma^* \to \Gamma^*$, where $\Sigma^*$ and $\Gamma^*$ denote the set of all possible finite sequences over the respective alphabets. The model, typically a transformer, is trained on a dataset $\mathcal{D}_{\text{train}}$ consisting of input-output pairs $(\boldsymbol{x}, \boldsymbol{y})$ where $\boldsymbol{x} \in \Sigma^*$, $\boldsymbol{y} = f(\boldsymbol{x})$, and the length of the input sequence $|\boldsymbol{x}| \le n$ for some maximum training length $n$.

Length generalization refers to the model's ability to perform well on input sequences $\boldsymbol{x}$ with lengths $N > n$ drawn from a test distribution $\mathcal{D}_{\text{test}}^N$. Formally, let $\mathcal{M}$ be a model trained on $\mathcal{D}_{\text{train}}$. Let $\text{Err}(\mathcal{M}, \mathcal{D})$ denote the expected error (i.e., $1 - \text{accuracy}$) of model $\mathcal{M}$ on distribution $\mathcal{D}$. We say that $\mathcal{M}$ exhibits *length generalization* if $\text{Err}(\mathcal{M}, \mathcal{D}_{\text{test}}^N)$ for $N > n$ remains low—close to $\text{Err}(M, \mathcal{D}_{\text{train}})$ for $N \approx n$ and degrading gracefully as $N$ increases—rather than exhibiting a sharp increase in error immediately beyond $n$. Length generalization contrasts with in-distribution generalization, which concerns performance on unseen examples $(\boldsymbol{x}, \boldsymbol{y})$ where $\boldsymbol{x} \sim \mathcal{D}_{\text{train}}$ and hence $|\boldsymbol{x}| \le n$.

In this work, we focus on decoder-only transformers. Let $\mathcal{M} = (E, D, K, Q, V)$ be a decoder-only single-layer transformer model with a single attention head and no MLP layer. Here $E$ and $D$ are the encoder and decoder matrices, and $K, Q, V$ are the key, query, and value matrices respectively.

## 2.1 Modeling Single-Token Outputs

We first consider the problem of computing a function $f : \Sigma^* \to \Gamma$ mapping each input sequence $\boldsymbol{x} = \langle x_1, \ldots, x_n \rangle$ to a *single* output symbol $f(\boldsymbol{x}) \in \Gamma$. We assume the existence of a special end-of-input symbol $\perp \notin \Sigma \cup \Gamma$. The transformer model $\mathcal{M}$ is given the augmented sequence $\boldsymbol{x} \circ \perp$, and we use the representation $z_\perp$ of the $\perp$ token after the attention layer to generate the output $\mathcal{M}(\boldsymbol{x}) = \arg\max_{j \in \Gamma} \langle z_\perp, \boldsymbol{d}_j \rangle$, where $\boldsymbol{d}_j$ is the decoder column corresponding to $j \in \Gamma$.

For $n \in \mathbb{Z}_+$, consider a distribution $\mathcal{D}_n$ on sequences of length $n$ from $\Sigma$. Let $\mathcal{D}$ refer to this ensemble of distributions $\mathcal{D}_n$, one for each $n \in \mathbb{Z}_+$.

**Definition 2.1** (Goodness). *A model $\mathcal{M}$ is $(\varepsilon, \gamma)$-good at length $n$ with respect to distribution $\mathcal{D}_n$ if*

$$\mathbb{P}_{\boldsymbol{x} \sim \mathcal{D}_n} \left[ \exists_{j \in \Gamma : j \ne f(\boldsymbol{x})} : \langle z_\perp, \boldsymbol{d}_j \rangle \ge \langle z_\perp, \boldsymbol{d}_{f(\boldsymbol{x})} \rangle - \gamma \right] \le \varepsilon,$$

*where $\boldsymbol{d}_j$ is the vector mapping to the output symbol $j \in \Gamma$. Here we call $\gamma$ the* margin *at length $n$.*

Observe that if the model is $(\varepsilon, \gamma)$-good for any value of $\gamma \ge 0$, then the probability of error is at most $\varepsilon$; moreover, a large value of $\gamma$ indicates that the model is correct *and* confident whp.

**Definition 2.2** (Sparse Functions). *A function $f : \Sigma^* \to \Gamma$ is $s$-sparse if for each $\boldsymbol{x} \in \Sigma^*$, there exists some set $A(\boldsymbol{x}) \subseteq \{1, \ldots, |\boldsymbol{x}|\}$ of $s$ positions such that for any subsequence $\boldsymbol{x}'$ of $\boldsymbol{x}$ containing the symbols at the positions in $A(\boldsymbol{x})$, we have $f(\boldsymbol{x}') = f(\boldsymbol{x})$. We call $A(\boldsymbol{x})$ the* anchor set *for $\boldsymbol{x}$.*

For example, the minimum and maximum functions are both 1-sparse functions, whereas the successor function (which, given an unsorted sequence $\boldsymbol{x}$, and a number $i$ in this sequence, outputs the next higher number in $\boldsymbol{x}$) is 3-sparse. We focus on $s$-sparse functions for the rest of this section.

**Definition 2.3** (Perturbation). *For $N > n$, define the $(N, n)$-perturbation of $\mathcal{D}_N$ as the distribution over sequences obtained by first drawing a random sequence $\boldsymbol{x}$ from $\mathcal{D}_N$, keeping the symbols in the anchor set $A(\boldsymbol{x})$ and keeping each symbol in the $N - s$ positions in $[N] \setminus A(\boldsymbol{x})$ independently with probability $\frac{n}{N-s} \ge \frac{n}{N}$, where $s = |A(\boldsymbol{x})|$. The expected length of the resulting sequence is $n + s$.*

We could have defined perturbations to be random sequences of length *exactly* $n + s$ obtained by retailing a uniformly random set of size $n$ from the $N - s$ positions; we focus on the i.i.d. case for simplicity of exposition. Note that in the typical case of $s \ll n$, the length of the subsequence obtained in the i.i.d. setting is $(1 \pm \varepsilon)(n + s)$ with probability $1 - \exp(-O(\varepsilon^2 n))$.

**Definition 2.4** (Robustness). *A model $\mathcal{M}$ is $(N, n)$-robust w.r.t. $\mathcal{D}$ with parameters $(\varepsilon, \gamma)$ if the model is $(\varepsilon, \gamma)$-good with respect to $\mathcal{D}_n$, and also with respect to the $(N, n)$-perturbation of $\mathcal{D}_N$.*

While our theoretical model considers the performance on sequences of length $\approx (1 \pm \varepsilon)(n+s)$ with high probability, our experiments are done on sequences of length $n + s$. This choice to consider i.i.d. sampling in our theoretical results is purely for analytical simplicity.

# 3   Length Generalization Bounds

We first consider the case of single-token outputs. Our main generalization bound here says that if the model is $(N, n)$-robust, then it is also good at length $N$ (with suitably weaker parameters). Formally, let $\|V\|$ denote the operator norm of the value matrix. Then we show:

**Theorem 3.1** (Single-Token Generalization). *For $c > 1$, let $\mathcal{M} = (E, D, K, Q, V)$ be a one-layer attention-only transformer; assume that the encoder matrix $E$ and decoder matrix $D$ have unit-length columns. Suppose $\mathcal{M}$ is $(N, n)$-robust with parameters $(\varepsilon, \gamma)$, where $\gamma > 2\|V\| \cdot (1 - 1/4c)$ for $N \leq cn$. Then $\mathcal{M}$ is $\left(O(\varepsilon \cdot N/n), 0\right)$-good at length $N$.*

In particular, the probability of erring at length $N$ degrades linearly in the sequence length. While the theorem does not depend on how the model is trained, or the length of sequences on which this training should be done, our experiments are performed when training on length up to $n + s$.

To prove Theorem 3.1, it will be convenient to have the following "anti-concentration" bound. (We use this only for the special case of independent random variables, but we prove it for a more general negative correlation property that captures sampling without replacement.)

**Theorem 3.2** (Anti-Concentration Bound). *Suppose $Z_1, \ldots, Z_m$ are identical Bernoulli random variables with $\mathbb{E}[Z_i] = q$ and satisfying the negative cylinder property that $\Pr[\wedge_{i \in T} Z_i = 0] \leq \prod_{i \in T} \Pr[Z_i = 0]$ for all $T \subseteq [m]$. Let $a_1, \ldots, a_m$ be non-negative real numbers and suppose $\sum_i a_i \geq 1$. Define $X = \sum_i a_i Z_i$ (hence $\mathbb{E}[X] \geq q$). Then, we have $\Pr[X \geq q/4] \geq q/2$.*

We can now return to the proof of our length generalization result for single-token outputs.

*Proof of Theorem 3.1.* Consider an input sequence $\boldsymbol{x} = \langle x_1, \ldots, x_N \rangle$ of length $N$ on which the model makes a mistake. Recall that we feed the augmented sequence $\boldsymbol{x} \circ \perp$ to the model, which returns $\arg\max_{j \in \Gamma} \langle z_\perp, \boldsymbol{d}_j \rangle$ as the answer. (Here $z_\perp$ is the embedding of the end-of-input token after passing through the model, and $\boldsymbol{d}_j$ is the column of the decoder matrix corresponding to output symbol $j \in \Gamma$.) Let $j^\star = f(\boldsymbol{x}) \in \Gamma$ be the correct answer on this sequence; since the model makes a mistake on $\boldsymbol{x}$, it outputs some other symbol $j^\dagger \in \Gamma, j^\dagger \neq j^\star$.

The attention structure of our model ensures that the representation of the $\perp$ token must be

$$z_\perp = \frac{\sum_{i=1}^N \exp(\langle K\boldsymbol{e}_{x_i}, Q\boldsymbol{e}_\perp \rangle) \cdot V\boldsymbol{e}_{x_i}}{\sum_{i=1}^N \exp(\langle K\boldsymbol{e}_{x_i}, Q\boldsymbol{e}_\perp \rangle)} = \frac{\sum_{i \in [N]} w_i \boldsymbol{v}_i}{\sum_{i \in [N]} w_i},$$

where $\boldsymbol{e}_{x_i}$ is the encoding of the symbol $x_i$ and $\boldsymbol{e}_\perp$ that of the end-of-input symbol $\perp$, and we use $\boldsymbol{v}_i = V\boldsymbol{e}_{x_i}$ and $w_i = \exp(\langle K\boldsymbol{e}_{x_i}, Q\boldsymbol{e}_\perp \rangle) \geq 0$ to reduce notation. Define $\alpha_i := \langle \boldsymbol{v}_i, \boldsymbol{d}_{j^\star} - \boldsymbol{d}_{j^\dagger} \rangle$. Since each of the $\boldsymbol{e}_i, \boldsymbol{d}_j$ vectors has unit length, we have the following upper bound on $\alpha_i$:

$$\alpha_i = \langle \boldsymbol{v}_i, \boldsymbol{d}_{j^\star} - \boldsymbol{d}_{j^\dagger} \rangle = \langle V\boldsymbol{e}_{x_i}, \boldsymbol{d}_{j^\star} - \boldsymbol{d}_{j^\dagger} \rangle \leq 2\|V\|. \tag{1}$$

Now, since the model errs on $\boldsymbol{x}$ and outputs $j^\dagger$ instead of $j^\star = f(\boldsymbol{x})$, we have

$$0 \geq \langle z_\perp, \boldsymbol{d}_{j^\star} - \boldsymbol{d}_{j^\dagger} \rangle = \frac{\sum_{i \in A(\boldsymbol{x})} w_i \alpha_i + \sum_{i \in [N] \setminus A(\boldsymbol{x})} w_i \alpha_i}{\sum_{i \in [N]} w_i}. \tag{2}$$

Suppose the model errs at length $N$ with probability $\delta$. We want to show the following claim: *conditioned on making an error (say on sequence $\boldsymbol{x}$), the probability that the margin is less than $\gamma$ on the $(N, n)$-perturbation of $\mathcal{D}$ is at least $\delta p/2$, where $p = \frac{n}{N-s} \geq \frac{n}{N} \geq \frac{1}{c}$. Indeed, by assumption, we know this probability of small margin must be smaller than $\varepsilon$, so $\delta \cdot n/(2N) \leq \delta p/2 \leq \varepsilon$, so $\delta \leq \varepsilon \cdot (2N/n)$, proving the theorem.

Now we prove the claim. Define $Q' := \{i \in [N] \setminus A(\boldsymbol{x}) \mid \alpha_i < 0\}$ to be the indices of the negative terms in the numerator of (2) but outside the anchor set and let $P' := [N] \setminus Q'$. $P'$ thus contains either the indices of the non-negative terms in the numerator of (2) or indices that are in the anchor set (hence retained in the perturbation with probability 1). We focus on this numerator and regroup terms to get:

$$\sum_{i \in P'} w_i \alpha_i \leq \sum_{i \in Q'} w_i |\alpha_i|. \tag{3}$$

Recall the definition of the $(N, n)$-perturbation of the sequence: it retains all the $s$ tokens that "force" the correct output $j^\star$ (i.e., samples them with probability 1) and samples each other token independently with probability $p$. Let $Y_i$ be the indicator r.v. for the token $\sigma_i$ with $i \in [N]$ being sampled, and let $\tilde{x}$ denote the (random) shorter subsequence obtained from $x$ by restricting to the sampled indices (and $\tilde{z}$ being the r.v. denoting the embedding of the $\perp$ token on this input). Consider the outcomes of $\{Y_i\}$ for $i \in P'$, and let us condition on their values being $Y_i = y_i$; this gives us

$$\langle \tilde{z}_\perp, \boldsymbol{d}_{j^\star} - \boldsymbol{d}_{j^\dagger} \rangle = \frac{\sum_{i \in P'} w_i \alpha_i y_i - \sum_{i \in Q'} w_i |\alpha_i| Y_i}{\sum_{i \in P'} w_i y_i + \sum_{i \in Q'} w_i Y_i}. \tag{4}$$

Now, we consider the sum $S := \sum_{i \in P'} w_i \alpha_i y_i$. If $S \leq 0$, then we are done since it means the model made an error at length $n$ and hence the small margin claim holds vacuously. Therefore, we focus on $S > 0$. Now define $a_i := w_i |\alpha_i| / S$ for $i \in Q'$; observe that $\sum_{i \in Q'} a_i \geq 1$ because of (3). Hence the $a_i$ values and the associated i.i.d. r.v.'s $Y_i$ satisfy the conditions of Theorem 3.2, which implies that $\Pr[\sum_{i \in Q'} w_i |\alpha_i| Y_i \geq Sp/4] \geq p/2$. Moreover, $\sum_{i \in Q'} w_i Y_i \geq 0$ always holds for any indicator r.v.'s $Y_i$ in $Q'$. Substituting these back into (4), we get that conditioned on $\mathcal{M}$ making a mistake on $x$, and for any conditioning $Y_i = y_i$ for the r.v.'s in $P'$, the following upper bound on the margin holds w.p. at least $p/2$:

$$\langle \tilde{z}_\perp, \boldsymbol{d}_{j^\star} - \boldsymbol{d}_{j^\dagger} \rangle \leq \frac{(1 - p/4) \cdot \sum_{i \in P'} w_i \alpha_i y_i}{\sum_{i \in P'} w_i y_i} \overset{(1)}{\leq} 2\|V\| \cdot \left(1 - \frac{p}{4}\right) \leq 2\|V\| \cdot \left(1 - \frac{1}{4c}\right). \quad \square$$

### 3.1 Extension to Multi-Token Outputs

Our theoretical framework also extends naturally to capture the more realistic scenario where the output is a sequence of tokens as opposed to a single token. Note that in this case one can view the problem of predicting each output token as an individual single-token output prediction problem with a corresponding auxiliary task. Hence, in the multi-token output setting we have a set of auxiliary tasks related to the main task, one per output position. Based on this we provide the following guarantee. We defer the full details to the Appendix.

**Theorem 3.3** (Multi-Token Generalization). *For $c > 1$, let $\mathcal{M} = (E, D, K, Q, V)$ be a one-layer attention-only transformer; assume that the encoder matrix $E$ and decoder matrix $D$ have unit-length columns. Suppose $\mathcal{M}$ is $(N, n)$-multi-token-robust with parameters $(\varepsilon, \gamma)$, where $\gamma > 2\|V\| \cdot (1 - 1/4c)$ for $N \leq cn$. Then $M$ is $\left(O(\varepsilon \cdot N/n), 0\right)$-good at length $N$.*

### 3.2 A Training Recipe for Length Generalization

Our framework suggests a natural training recipe for inducing length generalization for sparse tasks.

1. *Identifying auxiliary tasks:* For a sparse function, we produce multi-token outputs, where each token at timestep $t$ is obtained by applying some "local" function $f_t$ to a small number of previous tokens. Given the sparsity, there can only be a bounded number of such functions; moreover, for many common tasks, there are just a handful of functions being repeatedly applied, e.g., finding the minimum and successor in sorting, or doing single-digit addition in the increment task, or summing two tokens in the SLiM task, etc. These commonly-used functions for a particular problem are natural auxiliary tasks for it.
2. *Robust training for auxiliary tasks:* Having identified these auxiliary tasks, we train the model to be $(N, n)$-robust on auxiliary tasks (in addition to training on the main task). Specifically, we use the standard multi-task training where, during each gradient step, a batch of data from either the main task or an auxiliary task is used (in a round-robin manner) for backpropagation. Note that all training is on $n$-length sequences.
3. *Robustness implies length generalization:* We can now use our theoretical guarantees, which say that if we are $(N, n)$-robust with good parameters, then length generalization happens (i.e., low error on $N$-length sequences).

## 4 Experiments

In this section we present empirical results evaluating our robust training recipe for length generalization. The goal is to assess its effectiveness in improving length generalization on challenging

and some well-known synthetic tasks and to validate the underlying theoretical assumptions in our framework. We begin by describing the synthetic tasks that we consider, and the corresponding auxiliary tasks. In each case we will provide an example illustration of the main task and the corresponding auxiliary task. When presenting the auxiliary task we will use a red colored token to represent the current token for which the next token output needs to be found. We will use blue colored tokens to highlight the corresponding relevant tokens for the auxiliary task. (More details about the experimental setup and auxiliary tasks appear in Appendix C.)

## 4.1 Synthetic Tasks

**Sorting.** We consider sorting sequences of a particular length $m$ where each element is drawn uniformly at random from $\{1, \ldots, 100\}$. Given an input sequence $\boldsymbol{x} = \langle x_1, \ldots, x_m, \bot \rangle$ the main task corresponds to outputting the sorted sequence. The natural auxiliary task is to produce the successor of a given element in the output sequence. Figure 1 shows an example of the sorting task and a corresponding auxiliary task. For the sorting task we train on sequences of length up to 20 and measure length generalization for higher length sequences.

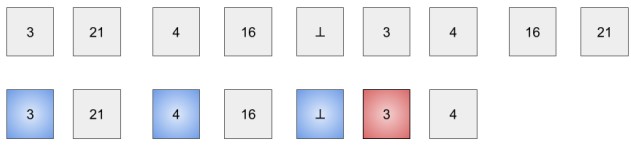

Figure 1: An example of the sorting main task and an auxiliary task.

**Reversal.** We consider the task of reversing a given sequence where each element is drawn uniformly at random from $\{1, \ldots, 100\}$. Note that for the reversal task, a model has to crucially keep track of the position in the input of the current token that is being output. We consider the following task representation that makes this explicit and lets us design auxiliary tasks that depend on a small number of relevant tokens: we explicitly append *pos* and *val* tokens to keep track of the position information; see Figure 2 for an example. We train the reversal task on sequences of length up to 20.

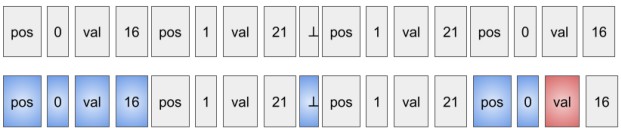

Figure 2: An example of the reversal main task and an auxiliary task: to output a value, we pay attention to the current position (which is 0), and to the corresponding pos 0 and val 16 in the input.

**Copy.** This is similar to the reversal task except that one is required to copy the input instead of reversing it; see Figure 3 for an example. We use similar representation as with reversal and generate training data for the main task and the auxiliary tasks for sequences up to length 20.

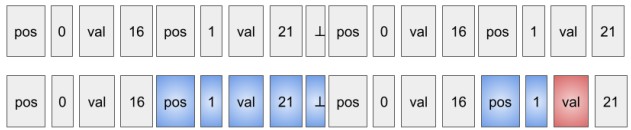

Figure 3: An example of the copy main task and an auxiliary task.

**Increment.** The increment task consists of adding 1 to a sequence of digits chosen from 0 to 9. As is standard for length generalization on arithmetic tasks, we consider producing the incremented number in reverse format (lowest significant digit first). Furthermore, we again include *pos* token along with the *sum* and *carry* tokens that let us define sparse auxiliary tasks; see Figure 4 for an example. We train the increment task for sequences of length up to 20.

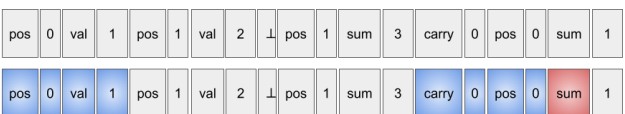

Figure 4: An example of the increment main task and an auxiliary task. Since we have a carry of $0$ at position $0$, and a value of $1$ at this position in the input, the output is $1$.

**Parity.** In this task given a sequence of $n$ bits we are required to compute $n$ parity bits where the $i$th output bit must equal the parity of the first $i$ bits of the input. Similar to the reversal and copy tasks, we also augment the training sequence with *pos* and *val* tokens to keep track of the current position information. Furthermore we also use the *xor* token to keep track of the parity of the prefix computed so far; see Figure 5 for an example. We generate training data for the main task and the auxiliary tasks for sequences up to length $20$.

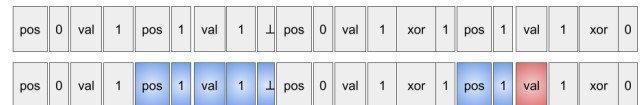

Figure 5: An example of the parity main task and an auxiliary task.

**SLiM (Small Language Modeling).** This synthetic task is inspired bylanguage modeling, and in particular, the notion of contextual lookup/recall. In a typical language modeling setting, each output token depends on a small number of prior tokens. However, these tokens may not always be the most recent ones, but may appear at arbitrary prior positions in the input sequence. Furthermore, given the correct tokens to attend to, the rule for producing the next output token is more involved than a simple copy operation. We consider an idealized scenario to capture this behavior. We consider the vocabulary $\{1, \ldots, 100\}$ that is randomly partitioned into two classes $A$ and $B$. The sequence is seeded with a set of four tokens, two chosen randomly from class A and the other two randomly from class B. Given a current output token from class A (class B, resp.), the next token is produced by finding the previous class A token (class B, resp.) in the sequence, and computing the sum of the two values (modulo 100). See Figure 6 for an example. Here, given an input sequence, followed by a "query" token $q$, the auxiliary task is to look up the last input token $r$ of the same type as $q$, and to output $q + r$ (modulo 100). We train the SLiM task on sequences of length up to 20.

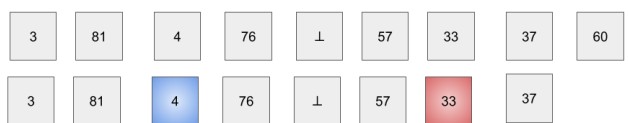

Figure 6: An example of the SLiM main task and an auxiliary task. In this example numbers from $1$ to $50$ belong to class A and the numbers from $51$ to $100$ belong to class B. The token $33$ pays attention to the previous token of the same class, i.e., to $4$, and sums to output $37$.

## 4.2 Training Details

A standard decoder-only transformer architecture is used (unless otherwise specified), with details on the number of layers, heads, embedding dimensions, and activation functions provided in the Appendix. We trained each model without positional embeddings. The proposed multi-task training method used the same backbone architecture with an additional output head for the auxiliary tasks. For all the tasks, we train on sequences of length up to $20$. In order to generate the training data for the main task, we sample a length $n$ from $[4, 20]$ at random and chose a random sequence of length $n$. (For the increment task we consider a slightly non-uniform sampling where $20\%$ of the training data consists of sequences that end with a $9$.) Crucially, the training data for the auxiliary tasks is generated from $(N, n)$-perturbations: i.e., first drawing an input sequence of a larger length and then random subsampling the irrelevant tokens down to length $n$. (As mentioned above, we subsample

these tokens without replacement to get a fixed length, instead of sampling each one independently.) The amount of training data used for the two approaches is kept equal, so that the baseline model uses the same amount of data as the proposed model. Moreover, both the baseline and proposed models use scratchpads; the baseline models do not length-generalize even with scratchpads.

Models were trained using the AdamW optimizer with a cosine learning rate schedule and a batch size of 1024 (see Appendix). The baseline model was trained solely on the primary task; our model was trained with auxiliary tasks using multi-task training. The primary metric was the "exact match" accuracy, i.e., the fraction of test sequences for which the model's output matched the ground truth.

## 4.3 Results

Figure 7 summarizes the results across the synthetic tasks that we consider. The baseline models, while achieving high accuracy within the training length distribution (length at most $n$), exhibit the typical sharp drop in performance for lengths beyond $n$. In contrast, the models trained with the auxiliary tasks maintain high accuracy far beyond $n$. For instance, on the sorting task, training on lengths up to 20, the baseline model's accuracy drops below $1\%$ for length 100, whereas our method achieves over $90\%$ accuracy. Similarly strong gains are observed for the other tasks as well.

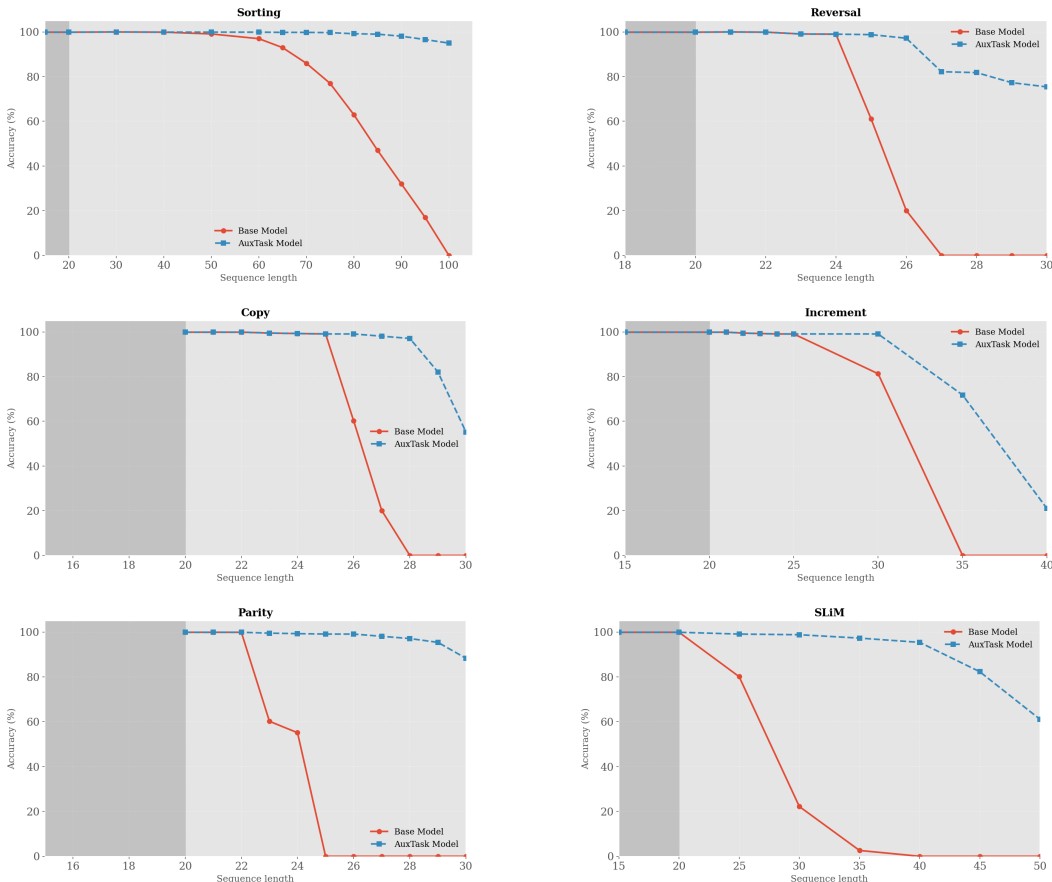

Figure 7: A comparison of the length generalization performance of the baseline training method vs. the proposed auxiliary task based method on various synthetic tasks. The $x$-axis represents the sequence length and the $y$-axis is the exact match accuracy. The shaded region represents the sequence lengths used during training and the non-shaded region represents the sequence lengths that the model does not see during training time.

We make a few observations about the results. Note that the performance on sorting is good up to length 60 even for the baseline model. This is consistent with similar observations made in prior works (Zhou et al., 2024a). The performance of the baseline model starts to drop sharply beyond

60 whereas the model trained with auxiliary tasks maintains high accuracy (more then 95%) even up to length 100. For tasks that involved carefully maintaining position information and indexing into a specific position in the input the performance of the baseline model drops sharply whereas the auxiliary task trained model has much greater robustness. Finally, the SLiM task, which involves learning simultaneously a rule for attending to specific tokens and a rule for using the attended tokens is significantly harder for the baseline model and its performance drops sharply beyond the training length. We again observe that by incorporating the auxiliary tasks in the training procedure the model is robust for up to two times the training length.

These results strongly support the hypothesis that explicitly training on the auxiliary tasks suggested by our framework, and using perturbed distributions, effectively guides the model to learn the internal mechanisms necessary for length generalization.

## 4.4   Scaling Experiments

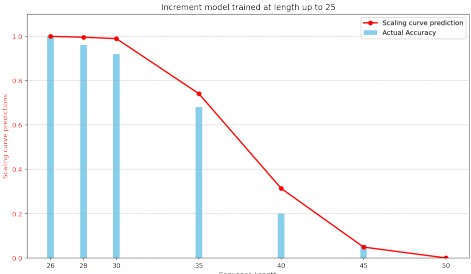 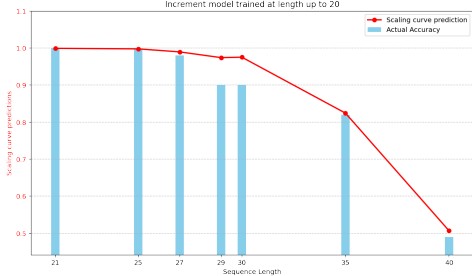

Figure 8: Scaling plots for the increment task. The left figure shows the actual accuracy at various lengths and the accuracy predicted by (5), for a model trained for length up to 20. The right figure shows the same plot for a model trained for length up to 25.

In this section we further aim to validate the practical relevance of our framework. Note that our key conceptual contribution is to establish a connection between the performance of a model on auxiliary tasks with its ability to length-generalize. Our main claim (Theorem 3.1) needs one to achieve a margin of $2\|V\|(1 - \frac{1}{4c})$ over the set of auxiliary tasks. However, this bound may be too pessimistic in real settings. In particular, analysis of a typical average case scenario reveals the following *rule-of-thumb* for length generalization:

$$\Pr\left[\text{error at length } N = c \cdot N\right] \sim \Pr\left[\text{aux. task margin } \leq \alpha\frac{(1 - 1/c)}{n}\right], \tag{5}$$

where $\alpha$ is a problem dependent constant.

We then perform a series of scaling experiments where we train models for our proposed synthetic tasks at various training lengths $n$ and compare how well does our proposed predicted length generalization performance from (5) match the empirically observed accuracy. Figure 8 shows one instance of this for the increment task trained at two different lengths (see Appendix for other scaling experiments). We see that (5) is consistently a good predictor of the length generalization performance across various values of $N$. These experiments help validate the connection between the theoretical assumptions made in our framework and achievable performance in practical settings.

## 5   Conclusion

Our work aims to bridge the gap between theoretical understanding and practical improvements in length generalization. By deriving auxiliary tasks directly from a novel framework, we offer a principled alternative to current mostly heuristic approaches. Limitations of our work include the need to further validate the assumptions and extend the framework to more complex tasks, particularly in natural language processing where defining clear algorithmic structures and relevant auxiliary tasks can be challenging. The specific theoretical models analyzed (e.g., single-layer transformers) are simplifications, and extending our results to deeper, more complex architectures is an important future step. Additional interesting directions include relaxing the theoretical assumptions and automatically discovering relevant auxiliary tasks.

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

# A  Extension to Multi-Token Prediction

In this section we extend our previous analysis to the broader case of functions that map an input sequence to an output sequence, rather than a single token. In particular, we consider the problem of computing a function $\hat{f} : \Sigma^* \mapsto \Gamma^*$ mapping each input sequence $\boldsymbol{x} = \langle x_1, \ldots, x_n \rangle$ to the output sequence $\tilde{\boldsymbol{x}} = \langle \tilde{x}_1, \ldots, \tilde{x}_m \rangle$, where $m := m(n)$ is the length of the output depending on $n$. As before we append the $\perp$ token to the input $\boldsymbol{x}$.

Note that as opposed to the single token case where we only have one output token conditioned on the embedding of the $\perp$ token, in the multi-token case we have $m > 1$ output tokens. Hence, for any function $f$ we can associate a sequence of single-token output functions $\langle f_1, \ldots, f_m \rangle$ where $f_1$ maps $\boldsymbol{x} \circ \perp$ to $\tilde{x}_1$ and $f_i$ maps $\boldsymbol{x} \circ \perp \circ \tilde{x}_1 \circ \cdots \circ \tilde{x}_{i-1}$ to $\tilde{x}_i$.

Analogously we will extend the definitions from Section 2 to the multi-token case. We will use $z_\perp$ to denote the (last layer) embedding of the $\perp$ token and $z_i$ to denote the embedding of the token $\tilde{x}_i$.

**Definition A.1** (Multi-Token Goodness). *A model $\mathcal{M}$ is $(\varepsilon, \gamma)$-good at length $n$ with respect to distribution $\mathcal{D}_n$ if*

$$\mathbb{P}_{\boldsymbol{x} \sim \mathcal{D}_n} \left[ \exists_{j \in \Gamma : j \neq \tilde{x}_1} : \langle z_\perp, \boldsymbol{d}_j \rangle \geq \langle z_\perp, \boldsymbol{d}_{\tilde{x}_1} \rangle - \gamma \right] \leq \varepsilon,$$

$$\mathbb{P}_{\boldsymbol{x} \sim \mathcal{D}_n} \left[ \exists i \geq 2, \exists_{j \in \Gamma : j \neq \tilde{x}_i} : \langle z_{i-1}, \boldsymbol{d}_j \rangle \geq \langle z_{i-1}, \boldsymbol{d}_{\tilde{x}_i} \rangle - \gamma \right] \leq \varepsilon,$$

*where $\boldsymbol{d}_j$ is the vector mapping to the output symbol $j \in \Gamma$. Here we call $\gamma$ the* margin *at length $n$.*

**Definition A.2** (Sparse Multi-Token Functions). *A function $f : \Sigma^* \to \Gamma^* = \langle f_1, \ldots, f_m \rangle$ is $s$-sparse if for each $i \in \{1, \ldots, m\}$, the function $f_i$ is $s$-sparse.*

**Definition A.3** (Perturbation Set). *For $N > n$, define the $(N, n)$-perturbation set of $\mathcal{D}_N$ as the product distribution $s_1 \times \cdots \times s_{m(N)}$ where each $s_i$ is an $(N + i, n)$-perturbation of $f_i$.*

We are now ready to define what it means for a transformer model for multi-token prediction to be robust.

**Definition A.4** (Multi-Token Robustness). *A model $\mathcal{M}$ is $(N, n)$-multi-token-robust w.r.t. $\mathcal{D}$ with parameters $(\varepsilon, \gamma)$ if the model is $(\varepsilon, \gamma)$-multi-token good with respect to $\mathcal{D}_n$, and also with respect to the $(N, n)$-perturbation set of $\mathcal{D}_N$.*

With the above definitions we have the following theorem.

**Theorem A.5** (Multi-Token Generalization). *For $c > 1$, let $\mathcal{M} = (E, D, K, Q, V)$ be a one-layer attention-only transformer; assume that the encoder matrix $E$ and decoder matrix $D$ have unit-length columns. Suppose $\mathcal{M}$ is $(N, n)$-multi-token-robust with parameters $(\varepsilon, \gamma)$, where $\gamma > 2\|V\| \cdot (1 - 1/4c)$ for $N \leq cn$. Then $\mathcal{M}$ is $\left( O(\varepsilon \cdot (N+m(N))/n), 0 \right)$-good at length $N$.*

*Proof.* The proof very closely mirrors the proof of the single token case. Consider an input sequence $\boldsymbol{x} = \langle x_1, \ldots, x_N \rangle$ of length $N$ on which the model makes a mistake. Let $p$ be the first output position where a mistake is made. In other words the model makes a mistake at $f_p$. Recall that we feed the augmented sequence $\boldsymbol{x} \circ \perp \circ \tilde{x}_1, \ldots, \tilde{x}_{p-1}$ to the model, which returns $\arg\max_{j \in \Gamma} \langle z_{p-1}, \boldsymbol{d}_j \rangle$ as the answer. (Here $z_{p-1}$ is the embedding of $\tilde{x}_{p-1}$ after passing through the model, and $\boldsymbol{d}_j$ is the column of the decoder matrix corresponding to output symbol $j \in \Gamma$.) Let $j^\star = \tilde{x}_p \in \Gamma$ be the correct answer on this sequence; since the model makes a mistake on $\boldsymbol{x}$, it outputs some other symbol $j^\dagger \in \Gamma, j^\dagger \neq j^\star$. Let $N' = N + p \leq N + m(N)$ be the length of the sequence that is input to the model.

The attention structure of our model ensures that the representation of $\tilde{x}_{p-1}$ token must be

$$z_{p-1} = \frac{\sum_{i=1}^{N'} \exp(\langle K\boldsymbol{e}_{x_i}, Q\boldsymbol{e}_{p-1} \rangle) \cdot V\boldsymbol{e}_{x_i}}{\sum_{i=1}^{N'} \exp(\langle K\boldsymbol{e}_{x_i}, Q\boldsymbol{e}_{p-1} \rangle)} = \frac{\sum_{i \in [N']} w_i \boldsymbol{v}_i}{\sum_{i \in [N']} w_i},$$

where $\boldsymbol{e}_{x_i}$ is the encoding of the symbol $x_i$ and $\boldsymbol{e}_{p-1}$ that of $\tilde{x}_{p-1}$, and we use $\boldsymbol{v}_i = V\boldsymbol{e}_{x_i}$ and $w_i = \exp(\langle K\boldsymbol{e}_{x_i}, Q\boldsymbol{e}_{p-1} \rangle) \geq 0$ to reduce notation. Define $\alpha_i := \langle \boldsymbol{v}_i, \boldsymbol{d}_{j^\star} - \boldsymbol{d}_{j^\dagger} \rangle$. Since each of the $\boldsymbol{e}_i, \boldsymbol{d}_j$ vectors has unit length, we have the following upper bound on $\alpha_i$:

$$\alpha_i = \langle \boldsymbol{v}_i, \boldsymbol{d}_{j^\star} - \boldsymbol{d}_{j^\dagger} \rangle = \langle V\boldsymbol{e}_{x_i}, \boldsymbol{d}_{j^\star} - \boldsymbol{d}_{j^\dagger} \rangle \leq 2\|V\|. \tag{6}$$

Now, since the model errs on $\boldsymbol{x}$ and outputs $j^\dagger$ instead of $j^\star = f(\boldsymbol{x})$, we have

$$0 \geq \langle z_{p-1}, \boldsymbol{d}_{j^\star} - \boldsymbol{d}_{j^\dagger} \rangle = \frac{\sum_{i \in A(\boldsymbol{x})} w_i \alpha_i + \sum_{i \in [N'] \setminus A(\boldsymbol{x})} w_i \alpha_i}{\sum_{i \in [N']} w_i}, \tag{7}$$

where $A(\boldsymbol{x})$ is the sparse set for the function $f_p$. Now, suppose the model errs at length $N'$ with probability $\delta$. We want to show the following claim: *conditioned on making an error (say on sequence $\boldsymbol{x}$), the probability that the margin is less than $\gamma$ on the $(N', n)$-perturbation of $\mathcal{D}$ is at least $\delta p / 2$, where $p = \frac{n}{N'-s} \geq \frac{n}{N'} \geq \frac{1}{c}$.* Indeed, by assumption, we know this probability of small margin must be smaller than $\varepsilon$, so $\delta \cdot n/(2N') \leq \delta p/2 \leq \varepsilon$, so $\delta \leq \varepsilon \cdot (2N'/n)$, proving the theorem.

Now we prove the claim. Define $Q' := \{i \in [N'] \setminus A(\boldsymbol{x}) \mid \alpha_i < 0\}$ to be the indices of the negative terms in the numerator of (7) but outside the anchor set and let $P' := [N'] \setminus Q'$. $P'$ thus contains either the indices of the non-negative terms in the numerator of (7) or indices that are in the anchor set (hence retained in the perturbation with probability 1). We focus on this numerator and regroup terms to get:

$$\sum_{i \in P'} w_i \alpha_i \leq \sum_{i \in Q'} w_i |\alpha_i|. \tag{8}$$

Recall the definition of the $(N', n)$-perturbation of the sequence: it retains all the $s$ tokens that "force" the correct output $j^\star$ (i.e., samples them with probability 1) and samples each other token independently with probability $p$. Let $Y_i$ be the indicator r.v. for the token $\sigma_i$ with $i \in [N']$ being sampled, and let $\hat{\boldsymbol{x}}$ denote the (random) shorter subsequence obtained from $\boldsymbol{x}$ by restricting to the sampled indices (and $\hat{z}_{p-1}$ being the r.v. denoting the embedding of $\tilde{x}_{p-1}$ token on this input). Consider the outcomes of $\{Y_i\}$ for $i \in P'$, and let us condition on their values being $Y_i = y_i$; this gives us

$$\langle \hat{z}_{p-1}, \boldsymbol{d}_{j^\star} - \boldsymbol{d}_{j^\dagger} \rangle = \frac{\sum_{i \in P'} w_i \alpha_i y_i - \sum_{i \in Q'} w_i |\alpha_i| Y_i}{\sum_{i \in P'} w_i y_i + \sum_{i \in Q'} w_i Y_i}. \tag{9}$$

Now, we consider the sum $S := \sum_{i \in P'} w_i \alpha_i y_i$. If $S \leq 0$, then we are done since it means the model made an error at length $n$ and hence the small margin claim holds vacuously. Therefore, we focus on $S > 0$. Now define $a_i := w_i |\alpha_i|/S$ for $i \in Q'$; observe that $\sum_{i \in Q'} a_i \geq 1$ because of (8). Hence the $a_i$ values and the associated i.i.d. random variables $Y_i$ satisfy the conditions of Theorem 3.2, which implies that $\Pr[\sum_{i \in Q'} w_i |\alpha_i| Y_i \geq Sp/4] \geq p/2$. Moreover, $\sum_{i \in Q'} w_i Y_i \geq 0$ always holds for any indicator r.v.'s $Y_i$ in $Q'$. Substituting these back into (9), we get that conditioned on $\mathcal{M}$ making a mistake on $\boldsymbol{x}$, and for any conditioning $Y_i = y_i$ for the r.v.'s in $P'$, the following upper bound on the margin holds w.p. at least $p/2$:

$$\langle \hat{z}_{p-1}, \boldsymbol{d}_{j^\star} - \boldsymbol{d}_{j^\dagger} \rangle \leq \frac{(1-p/4) \cdot \sum_{i \in P'} w_i \alpha_i y_i}{\sum_{i \in P'} w_i y_i} \overset{(6)}{\leq} 2\|V\| \cdot \left(1 - \frac{p}{4}\right) \leq 2\|V\| \cdot \left(1 - \frac{1}{4c}\right). \quad \square$$

## B  Proof of Theorem 3.2

*Proof of Theorem 3.2.* By scaling the $a_i$'s down if needed, let us assume that $\sum_i a_i = 1$, so that $\mathbb{E}[X] = q$. There are two cases:

1. Suppose there exists $i'$ such that $a_{i'} \geq q/4$. Then
$$\Pr[X \geq q/4] \geq \Pr[a_{i'} Z_{i'} \geq q/4] = \Pr[Z_{i'} = 1] = q \geq q/2.$$

2. Else, each $a_i \leq q/4$. Then define $a_i' = \frac{a_i}{q/4}$, $X_i' = a_i' Z_i$, and $X' = \sum_i X_i'$. Now we have $a_i' \in [0, 1]$ and also $X_i' \in [0, 1]$, but $\mathbb{E}[\sum_i X_i'] = 4$. We want to bound show that $\Pr[X' \geq 1]$ is large. Indeed, we can use a concentration bound for negative cylinder random variables (Dubhashi and Panconesi, 2009; Panconesi and Srinivasan, 1997) to show that the probability of a "bad" event is:

$$\Pr[X' \leq 1] = \Pr\left[X' \leq \mathbb{E}[X']/4\right]$$

$$\leq \exp\left\{-\frac{\mathbb{E}[X'] \cdot (1 - 1/4)^2}{2}\right\} = e^{-(4-1)^2/(2\cdot 4)} \leq 1/e.$$

Hence,

$$\Pr[X \geq \mathbb{E}[X]/4] = \Pr[X' \geq \mathbb{E}[X']/4] \geq 1 - 1/e \geq q/2$$

for any $q \in [0, 1]$, as claimed.

Ths completes the proof of Theorem 3.2. □

## C   Experiment Details

All our experiments are conducted using the Jax Flaxformer codebase and involve training decoder only transformer models from scratch. For training the baseline models, we partition the training dataset for the main task into batches of a specified size (see details below) and use the AdamW optimizer. We run our models for a prespecified number of gradient steps. To set the learning rate, we start with a learning rate of zero and linearly ramp up to a prespecified base learning rate over 10 epochs. Following that we anneal the learning rate to zero using the standard cosine learning rate scheduler (with one cycle); see Table 1. We train all our models for $200,000$ gradient steps (the in-distribution performance saturates much before that for all the tasks). Besides the four main tasks mentioned in the main body, we also experiment with two additional tasks: the task of copying a sequence and the task of computing the parity of a given binary sequence.

| Parameter | Value |
|---:|:---|
| Embedding size ($d$) | 1024 |
| Vocabulary size ($q$) | 103 (for sorting, SLiM) 
 200 (for reversal, increment, copy and parity) |
| Position embedding type | None |
| # Attention heads ($h$) | 16 |
| MLP inner dimensionality ($d'$) | 2048 |
| Sequence length | 512 |
| Base learning rate | 1e-5 |
| Optimizer | AdamW |
| LR warmup | Linear for 10 epochs |
| LR decay schedule | Cosine, one cycle with default parameters |
| Dropout | None |
| Activation | GELU |
| Depth | 2 (for sorting) 
 4 (for reversal, copy, increment, parity, and SLiM) |

Table 1: Hyperparameters for the experiments.

## D   Dataset Details

Below we describe our example generation process for each of the synthetic tasks, for both main and auxiliary tasks. For the main task, we generate 100M examples, except for SLiM, where we generate 1M examples. For all synthetic tasks, we generate 100M examples for the auxiliary task.

### D.1   Sorting

For the main task we first pick a length value $\ell$ uniformly at random in the range $[4, 20]$. Given $\ell$, with probability $0.8$ we generate a length $\ell$ sequence by picking each of the $\ell$ numbers i.i.d from the vocabulary $V = \{1, \ldots, 100\}$. With probability $0.2$ we generate the sequence from a shortened vocabulary $V' \subset V$, a random subset of size $\ell/2$. This ensures that we train the models to do well even on inputs with repetitions.

To generate a training data point for the auxiliary task, we simply generate a sequence of length $N = 100$ at random following the same process as above and then pick an output position $i$ uniformly at random. We fix the relevant tokens needed to predict the token following position $i$ and subsample the rest with probability $n/N$.

### D.2 Reversal

For the main task we first pick a length value $\ell$ uniformly at random in the range $[4, 20]$. Given $\ell$, we generate a length $\ell$ sequence by picking the $\ell$ numbers i.i.d from the vocabulary $\{0, 1, \ldots, 100\}$.

To generate a training data point for the auxiliary task we simply generate a sequence of length $N = 60$ at random following the same process as above and then pick an output position $i$ uniformly at random. We fix the relevant tokens needed to predict the token following position $i$ and subsample with probability $n/N$. We also augment the training sequence with *pos* and *val* tokens to keep track of the current position information. In order for the models to be able to handle lengths larger than the ones seen during training, we generate *pos* values by sampling a position uniformly at random in the range $[0, 80]$ and incrementing it by one for each new digit in the sequence.

### D.3 Copy

For the main task we first pick a length value $\ell$ uniformly at random in the range $[4, 20]$. Given $\ell$, we generate a length $\ell$ sequence by picking the $\ell$ numbers i.i.d from the vocabulary $\{0, 1, \ldots, 100\}$.

To generate a training data point for the auxiliary task, we follow the same process as in reversal.

### D.4 Increment

For the main task we first pick a length value $\ell$ uniformly at random in the range $[4, 20]$. Given $\ell$, with probability $0.8$ we generate a length $\ell$ sequence by picking the number at the most significant position uniformly at random from $\{1, \ldots, 9\}$ and each of the remaining $\ell - 1$ numbers i.i.d from the vocabulary $V = \{0, \ldots, 9\}$. With probability $0.2$ we generate a random sequence that ends with a suffix of 9's where the suffix length is chosen to be uniform between $[1, \ell/2]$. This ensures that the models learn the *carry* operation faithfully.

To generate a training data point for the auxiliary task, we simply generate a sequence of length $N = 60$ at random following the same process as above and then pick an output position $i$ uniformly at random. We fix the relevant tokens needed to predict the token following position $i$ and subsample the rest with probability $n/N$. We also augment the training sequence with *pos*, *val*, *sum*, and *carry* tokens to keep track of the current position information. In order for the models to be able to handle lengths larger than the ones seen during training, we generate *pos* values by sampling a position at random in the range $[0, 80]$ and incrementing it by one for each new digit in the sequence. We also follow the standard practice of producing the output in reverse order, i.e., least significant digit first.

### D.5 Parity

For the main task we first pick a length value $\ell$ uniformly at random in the range $[4, 20]$. Given $\ell$, we generate a length $\ell$ sequence by picking the $\ell$ numbers i.i.d from the vocabulary $\{0, 1\}$. The output sequence corresponds to the sequential parity of every prefix of the input.

To generate a training data point for the auxiliary task, we simply generate a sequence of length $N = 60$ at random following the same process as above and then pick an output position $i$ uniformly at random. We fix the relevant tokens needed to predict the token following position $i$ and subsample with probability $n/N$. In order for the models to be able to handle lengths larger than the ones seen during training, we generate *pos* values by sampling a position uniformly at random in the range $[0, 80]$ and incrementing it by one for each new digit in the sequence.

### D.6 SLiM

For the main task we first pick a length $\ell$ uniformly at random in $[4, 20]$. We pick a seed sequence of length 4 by picking two random tokens from class $A$ and two from class $B$. Given the seed sequence, we generate a length-$\ell$ output sequence by applying the latent (deterministic) rule.

To generate a training data point for the auxiliary task we simply generate a sequence of length $N = 60$ at random following the same process as above and then pick an output position $i$ uniformly at random. We fix the relevant tokens needed to predict the token following position $i$ and subsample with probability $n/N$. So if the token at position $i$ is of type $A$, the auxiliary task is to find the previous token of type $A$ before position $i$, and output the sum of those two modulo 100.

