# OpenReview forum: "Length Generalization via Auxiliary Tasks"
_NeurIPS.cc/2025/Conference — NeurIPS 2025 poster_

### Official Review · Reviewer_1jdG · 2025-06-21

**Clarity:** 3
**Significance:** 3
**Originality:** 3
**Rating:** 3
**Confidence:** 4

**Summary:**

The paper focusses on a specialised set of sparse task setups, where the output token only depends on a small set of preceding tokens and studies these kind of tasks with a novel framework that enables length generalisation. The paper shows how good performance on smaller training lengths can lead to generalisation for such sparse tasks. For transformers, the error grows linearly in sequence length of the input. Thus that naturally leads to a recipe for training to help in length generalisation, which is to train on the native distribution of $\mathcal{D}_n$ and also on random n-length restrictions of $\mathcal{D}_N$ (the random restrictions here implies -- making the longer input with $N$ tokens smaller to length $n$, where all the $n$ tokens will be utilised to make the next prediction). This is the special key step being proposed here, as just training on $\mathcal{D}_n$ is already known to not lead to strong length generalisation. So, if I understand correctly, then the idea is to train both on sparse as well as in a certain sense dense setups [referred to auxiliary tasks here] at smaller training lengths, and that should lead to length generalisation for the sparse setting at longer lengths. Finally several experiments are conducted to verify the theoretical claims and show the training recipe being successfully applied to a variety of tasks.

**Questions:**

This is not a question, more of a suggestion / comment. Currently the paper writing seems to be written in 2 styles. The empirical experiments are quite easy to understand, insights are also intuitive and is quite well written. The theory on the other hand while is quite trivial to understand once one gets past the notation, I feel there is room to make the things a bit more intuitive for the reader as well. There can be for instance a paragraph defining the proof sketch, before the actual proof starts, to help prepare the reader a bit better for how the proof is going to go. However, this is not quite a criticism, just an observation, as the experimental section was quite intuitive, but the theory took a bit of time to go through because of the definitions and heavy notation, which could be circumvented with some kind of short proof sketches.


Overall I feel that this paper has quite a lot of potential and I really like the overall idea being proposed here. However, there needs to be some more interaction with prior literature, not just in terms of mentioning what has been proposed before, but in terms of how good the current method is compared to other things. Some more discussions about positional encodings, it's role and bit more exposition on how to actually identify the underlying auxiliary tasks in a general sense would be helpful. In case, auxiliary tasks cannot be found across setups, then that has to acknowledged in the paper somewhere.

**Ethical Concerns:**

["NO or VERY MINOR ethics concerns only"]

**Final Justification:**

The authors were able to clarify my questions. I like the proposed method and see considerable potential for the paper to improve through deeper engagement with prior work, experiments on other positional encodings, and a clearer delineation of where the method is applicable and where it is not. In light of the authors’ responses and the other reviews—particularly regarding the mention of Awasthi and Gupta on task hinting (as the proposed method is quite similar, and so therefore the current paper by the authors requires somewhat more contribution to be valuable contribution and not just a marginal one)—Therefore I feel the current version is not ready for publication, and needs some revisions to be a solid paper.

**Limitations:**

yes

**Quality:**

3

**Strengths And Weaknesses:**

**Strengths**
    - There is a clear practical insight that can be taken from theoretical findings, and a training recipe that can be applied readily.
    - Empirical results shown in the paper support the theoretical claims being made in the paper.
    - The core idea of having an auxiliary task (reduce the actual task by reducing the number of tokens to the ones that matter and that don't change the answer) and then train based on it seems quite useful and is grounds for further exploration on its own.

**Weaknesses**
  - Only No Positional Encodings have been considered. It's unclear what the inclusion of positional encodings will do, especially since, something like Absolute positional encodings requires longer positions to be also trained, so clipping the length to smaller lengths for the auxiliary tasks might make training slower. Similarly with Rotary positional encodings, the rotation angle etc. might require modifications to the training recipe proposed here.
- Not all tasks that we can think of have an obvious auxilliary task that we can train them on. For example, math problems don’t have obvious auxilliary tasks that I can think of, thus although the insights from the paper are actionable, they are not easily applicable beyond across settings.
- Empirical Comparison has only been made with the base model of their choice, discussion about other hints / tricks suggested by prior literature is not discussed. Thus, although training this way can be considered better than pure vanilla training, but it’s not clear, whether this is objective better than something like index hinting, providing scratch pads to the model etc.  For instance it was pointed out in the related section of the text -- that scratchpads are not universally effective, but neither is this approach as far as I see it. Thus for a fair comparison, there needs to be some kind of comparisons with scratchpads. Similar arguments can be made for other kinds of training recipes proposed in the literature before.

---

> ### Author Rebuttal · Authors · 2025-07-31
>
> We thank the reviewer for their suggestion, encouragement, and comments. We are happy that you appreciated the clean idea behind our work: that sparsity naturally suggests a set of “core” tasks, which can then be trained on robustly, and this can ensure length generalization. We hope that this conceptual simplicity/message will be useful more broadly, beyond the specific problems in this paper.
>
> Thank you also for the detailed and valuable suggestion: we will rephrase to present the proofs more transparently, and will also unify the presentation styles. We completely agree that it will make the ideas of the paper more accessible, and will help the reader. We will also add more connections and comparisons to previous work.
>
> Please find responses and clarifications to the other questions and comments below.
>
> **The Role of Encodings:** Thanks for the question. We view the role of position encodings as complementary (and indeed, additive) to our proposed recipe. Typically in order to train a model with positional encoding for length generalization standard techniques such as a random position offset is used during training. If positional encodings are present, the same techniques will apply to our framework too.
>
> **Choice of Auxiliary Tasks:** It is a great question to design auxiliary tasks automatically for more real world settings such as natural language tasks and math tasks. One way would be to have a “library” of “basis” functions for the “local” tasks, and then use these as subroutines in a general training setup where they can hopefully enable length generalization for a variety of downstream applications.
>
> **Comparison to other Models:** Regarding the use of *scratchpads*, we would like to note that in all our experiments the baseline models use scratchpads and still do not length-generalize; this is consistent with what other literature on this topic has observed. *Index hinting* techniques such as position coupling are quite powerful for length generalization but rely on the right hints to be fed into the model at inference time as well. These are very strong hints; our focus was to make the model naturally learn to attend to the right tokens via robust training without such direct hints, since such hints will typically not be available at inference time. We are happy to include these methods for comparison in the next version of the paper.

---

> ### Comment · Reviewer_1jdG · 2025-08-01
>
> I thank the authors for their response. I am not quite sure, I am fully following the responses made by the authors and would like more details from the authors.
>
> Regarding the choice of encodings for example. My question about them was motivated by the fact that I don't quite believe that the integration of standard techniques (like random offset for APE) for example would then retain the computational efficiency that the authors are hoping to achieve with their proposed approach. It would be good if the authors can provide a bit more intuition about why they think, that the same techniques as and when applied to standard training would also be applicable with their methods, and not loose out on the efficiency of their suggested approach. With APE and the positional offset trick for example, it takes quite a lot more samples at various positions (such that most positions are seed during training), to be able to achieve length generalization. Since that overhead is already quite high, would the approach suggested by authors still have a significant increase in efficiency while trying to achieve length generalization. The answer to this question highly impacts the overall contribution of the authors work. This is precisely why I would like to request the authors to engage a bit more in trying to explain their reasoning.
>
> For the choice of Auxiliary tasks for realistic settings, could the authors provide a simple example of how they foresee having a library of basis functions, this currently seems very abstract to me, and I am not sure I quite follow what the authors mean here.
>
> For the comparison to other models, I gave the paper a quick skim through again, to me, it was not clear where this was mentioned, that the baseline method already uses scratchpads. I would request the authors to provide a bit more details about how that baseline training was done (or kindly point me to a section in the main paper / appendix where more details are already provided), as that is quite relevant to understand the experimental setup and the significance of the results. I also do not quite understand why index hinting would not be available at inference time. It can always be fed. I understand that index hinting is costly (which is precisely why a comparison would be useful), but I don't quite understand why you cannot design the training in a way where models learn to use index hints during training and then during inference also (conditioned on the input being fed with the index hints), then use the same to solve the task. For example in Zhou et al [1], they were able to deploy index hints quite well, but maybe I am missing some details that make it hard to apply in general.
>
> Additionally I would like to acknowledge that I was not aware of the paper Awasthi and Gupta [2], which I only have become aware of because of reviewer hBbS, and the authors response to the same. I would like to thank both the reviewer and the authors for the same. While I acknowledge that there are certain additional improvements in the current work from that paper, the originality of the work is severely impacted by the fact there is a significant overlap with [AG23]. In my head, the strength of the paper is slightly weakened by the extent of the overlap, despite there being improvements.
>
> [1] Zhou et al, What algorithms can transformers learn ?
> [2] Awasthi and Gupta, Improving Length-Generalization in Transformers via Task Hinting

---

> > ### Author Response · Authors · 2025-08-04
> > **response to reviewer**
> >
> > **On positional encodings**
> >
> > Sorry, we misunderstood your original question: thank you for clarifying. If we understand correctly, you are worried that training on shorter lengths will require more training for the model to adapt to the particular positional encoding (say APE with random offsets) being used. We see your concern; however, we still feel our methods will efficiently compose, since the number of samples required to learn the patterns in the positional encoding are not likely to be the bottleneck here. However, we don’t have experiments or concrete evidence at this time to back our claims up.
> >
> > **On auxiliary tasks**
> >
> > If we were interested in math reasoning, the basis may consist of a toolkit of mathematical primitives: addition, multiplication, elementary functions, applying induction, and so on, with the hope that each of these primitives would be sparse and hence could be trained for.
> >
> > **On scratchpads**
> >
> > In our experiments given in Section 4, we describe how the main task and the auxiliary tasks are designed for each problem. The main task is what the baseline method is trained on and they all use scratchpads to solve the problem. For instance the increment problem involves keeping track of intermediate carry digits.
> >
> > **On index hints**
> >
> > Indeed, we meant that index hints and position coupling and such ideas require us to manipulate the input during *both* training and inference: as you say, the input needs to be fed with index hints. Our aim is to develop techniques where the input does _not_ have to be changed (or carefully formatted, with index hints added) at inference time, which is clearly desirable.

---

> > > ### Comment · Reviewer_1jdG · 2025-08-05
> > >
> > > I thank the authors once again for their response.
> > >
> > > - Scratchpad usage: My concerns about how the scratchpad was used in the experiments have been resolved. I suggest adding a brief sentence to the paper that makes this point explicit, as it was not obvious to me, where scatchpads were available or they weren't, based on the current version of the paper.
> > >
> > > - Index hints: I understand the authors’ motivation to study techniques that do not modify the input. However, the claims about the superiority of the proposed method should be phrased more cautiously. From the clarifications provided, existing approaches such as index hints are not inferior—but simply different from what the authors want to study. This nuance should be stated more clearly in the manuscript.
> > >
> > > - Auxiliary tasks: Some of the primitives mentioned by the authors for math tasks (for instance)—e.g., addition and multiplication—are not themselves provably length-generalizable. If an auxiliary task lacks length generalizability yet is required as a primitive, the main task is also unlikely to generalize. Hence, the scope of the proposed method is narrower than currently presented. I encourage the authors to temper their claims about its utility.
> > >
> > > -  Finally, regarding positional encodings, I am glad that the authors understood the question. I thank the authors for their response. I understand that doing such experiments would require time and effort. But I would again urge the authors to acknowledge this in the paper somewhere the fact these experiments would be a very critical extension that would be required in future to make the proposed method more applicable to real world LLMs, using other kinds of PEs (RoPE, APE etc.)
> > >
> > > Overall, my questions have been clarified. I like the proposed method and see considerable potential for the paper to improve through deeper engagement with prior work, experiments on other positional encodings, and a clearer delineation of where the method is applicable and where it is not. In light of the authors’ responses and the other reviews—particularly regarding the mention of Awasthi and Gupta on task hinting (as the proposed method is quite similar, and so therefore the current paper by the authors requires somewhat more contribution to be valuable contribution and not just a marginal one)—I will maintain my score. I do hope that the current concerns and feedback doesn't discourage the authors, and on the other hand provides actionable future directions to make the work stronger for future publications.

---

### Official Review · Reviewer_TSCa · 2025-07-01

**Clarity:** 2
**Significance:** 2
**Originality:** 3
**Rating:** 4
**Confidence:** 4

**Summary:**

This paper addresses length generalization, a major challenge facing transformers. It shows that length generalization can be boosted by adding training data from an auxiliary tasks at longer lengths.

**Questions:**

See Weaknesses, especially the definition of the Auxiliary Tasks and the link between the experiments and the theorem.

**Ethical Concerns:**

["NO or VERY MINOR ethics concerns only"]

**Final Justification:**

Based on the rebuttal, I trust the authors that they can address the Weaknesses that I raised. I have thus increased my score.

**Limitations:**

yes

**Quality:**

2

**Strengths And Weaknesses:**

Strengths:
* addresses a key problem facing transformers
* includes both theoretical analysis and experiments
* experiments show clear promise of the proposed method

Weaknesses:
* The paper claims that the auxiliary tasks are derived from the novel framework of theoretical analysis (e.g. line 343), but I didn’t follow the link. Section 3.2 is quite unspecific and informal. The link to Theorems 3.1, 3.3 remains unspecified.
* I found the definitions of the auxiliary tasks confusingly presented. Example: line 261: “produce the successor of a given element in a sequence” What exactly does this mean? I don’t understand the link to Figure 1. I presume the second row in the figure is the auxiliary task? But what exactly is the task? I had similar trouble understanding the other tasks.
* While I did understand the SLiM task, I didn’t understand how it is a plausible model of language modeling. It is both very simplified and quite contrived. Is there a theoretical or conceptual justification for it?

---

> ### Author Rebuttal · Authors · 2025-07-31
>
> We thank the reviewer for their suggestions and comments. We are happy that you concur with the fact that this is a key problem on which progress is essential, and that you also see the promise of our approaches to getting better length generalization.
>
> Please find responses and clarifications to your specific questions and comments below.
>
> **Link between Theoretical Results and Training Methodology**: Recall that we  focus on sparse functions. For this we consider $(N,n)$-perturbations (which are sequences of length $n$ produced by suitably sub-sampling $N$-length sequences), and train for both the auxiliary and main tasks on these sequences. Given a problem, we do the following
>
>    a. **Identifying Auxiliary Tasks**: For sparse problems with multi-token outputs, each token at timestep $t$ is obtained by applying some “local” function $f_t$ to a small number of previous tokens. Given this sparsity, there can only be a _bounded_ number of such functions; moreover, for many common tasks, there are just a handful of functions being repeatedly applied, e.g., finding the minimum and successor in sorting, or doing single-digit addition in the increment task, or summing two tokens in the SLiM task, etc. These commonly-used functions for a particular problem are natural auxiliary tasks for that problem.
>
>    b. **Robust Training for Auxiliary Tasks** : Having identified these auxiliary tasks, we train the model to be $(N,n)$-robust on auxiliary tasks (in addition to training on the main task). Recall that this training is on $n$-length sequences.
>
>
>    c. **Robustness implies Length Generalization**: We can now use our theoretical guarantees, which say that if we are $(N,n)$-robust with good parameters, then length generalization happens (i.e., low error on $N$-length sequences).
>
> We hope this clarifies the setup. We will make this more explicit in the next version.
>
> **Clarification of Sorting Task:** For sorting, the auxiliary task is as follows: given an input unordered sequence $S$ of tokens (which in our case are numbers modulo 100), and a specific “query” token $q$, the auxiliary task is to lookup the smallest number in $S$ that is greater than $q$ (“successor”). Note that this “lookup” task is precisely what sorting repeatedly performs, since it outputs the minimum element, and then repeatedly outputs the successor of the last-output token. We will clarify this in the revision.
>
>
> **Motivation for the SLiM Task:** The motivation for the SLiM task was to serve as a simple example of _contextual recall_, where each number pays attention to the closest previous number of the same “type” and sums the two. This gives us a precisely defined task on which we can objectively measure length generalization, while capturing some of the essential properties of language (i.e., contextual lookup and then some transformation of the tokens looked up this way).  One could consider more sophisticated probabilistic versions, e.g., each token $a_t$ pays attention to multiple previous tokens having some types (that depends on $a_t$’s type), and then outputs the next token chosen from a probability distribution.  This is another step closer to natural language and we leave this for future work.

---

> > ### Comment · Reviewer_TSCa · 2025-08-05
> >
> > Thanks to the authors for the response, which helps. I'll get back if I have more questions.

---

### Official Review · Reviewer_hBbS · 2025-07-03

**Clarity:** 3
**Significance:** 2
**Originality:** 1
**Rating:** 3
**Confidence:** 4

**Summary:**

This work addresses the problem of length generalization in small transformers from both a theoretical and empirical perspective. It argues that training on short sequences often leads models to rely on length-specific heuristics, failing to generalize to longer inputs. To mitigate this, the authors propose joint training on a main algorithmic task (e.g., sorting) and auxiliary tasks that isolate sparse, local subcomputations (e.g., predicting the successor of an element in sorted order). Their theoretical contribution establishes generalization bounds that grow linearly with input length, under a sparsity condition where each output token depends on a limited number of input positions. Experiments on structured synthetic tasks (sorting, reversal, Increment and small language modeling) show that this method enables generalization to inputs up to 10× longer than those seen during training, while standard training fails catastrophically. However, its application is limited to tasks with known structure, and real-world applicability remains untested.

**Questions:**

# Questions

1. For tasks like language modeling or open-domain QA, where the latent task structure is not known, how might one design or discover auxiliary tasks automatically?
2. Is there any experimental comparison to curriculum learning or random length scaling, which might also improve extrapolation by varying sequence lengths during training?

**Ethical Concerns:**

["NO or VERY MINOR ethics concerns only"]

**Final Justification:**

Significant overlap with prior work makes its core idea not novel. Connection to larger-scale models is not clear either. Authors rebuttal did not address those satisfactorily. I maintain my original score.

**Limitations:**

Yes

**Quality:**

2

**Strengths And Weaknesses:**

# Strengths
1. Theoretical Grounding: The paper offers a rigorous generalization bound for sparse sequence-to-sequence functions showing test error grows linearly as a function of length.
2. On a few synthetic tasks, jointly training a main task with an auxiliary task (that contains subcomputation) in a multitasking manner  enables length generalization where baselines collapse.

# Weaknesses

1. Significant overlap with Prior Work: The idea of improving length generalization through auxiliary task multitasking has clear precedent in "Improving Length-Generalization in Transformers via Task Hinting" (Chughtai et al., 2023), which uses similar tasks (sorting + successor) and shows that multitask training of a main algirthimc task with a relevant auxiliary task, say successor finding, significantly improves length generalization.                                                                                                                                                                                                                             This major overlap is not acknowledged (as far as reviewer can see) and limits the originality of the contribution.
2. Synthetic-Only Evaluation: All experiments are on clean, synthetic tasks with known structure. The paper does not attempt even lightweight transfer to real-world tasks or semi-structured inputs, leaving open questions about it practical relevance.
3. Manual Auxiliary Design: The auxiliary tasks require prior knowledge of the task structure (e.g., that sorting can be decomposed into comparisons or successors). It's unclear how to design such tasks in domains where the compositional structure is not obvious.
4. Missing comparison to alternative methods such as curriculum learning, or architectural constraints, which have also been shown to improve length generalization.

---

> ### Author Rebuttal · Authors · 2025-07-31
>
> We thank the reviewer for their encouragement and comments. We are glad that you liked the theoretical justification of the training methodology we propose, and that the experiments showed significant gains when baseline training methods failed.
>
> Please find responses and clarifications to your specific questions and comments below.
>
> **Relationship to Chugtai et al:** We were unable to locate a 2023 paper by "Chugtai et al." on this topic and suspect the reviewer may be referring to Awasthi and Gupta [AG23], given the similar problem space. We sincerely apologize for this oversight in our initial submission and will, of course, add a detailed comparison to our Related Work section.
>
> Our work presents three strict improvements over [AG23]:
>
> 1. **Theoretical Justification:** In [AG23], there was no formalization of why task hinting would lead to length generalization. We, on the other hand, provide a theoretical framework to show why auxiliary tasks help with length generalization.
>
> 2. **Theory suggests new Training Recipe:** Our theoretical analysis leads to a robust training recipe, which was not present in [AG23].
>
> 3. **Improved Performance via Robust Training:** The new robust training recipe leads to better empirical performance across a variety of tasks. For example, for the increment task [AG23] length-generalized (i.e., obtain more than 90% test accuracy) _only_ by a factor of at most 1.2.  In our work, however, we can length generalize to a factor of more than 1.5 for the same task.
>
> **Use of Synthetic Tasks:** Synthetic tasks are standard in the literature on length generalization as it is easy to clearly define the task for various input lengths ($n$). Moreover, we wanted to focus the investigation on tasks with precise answers, like sorting, increment, etc., where the effectiveness of length generalization is concretely and objectively measurable. In fact, the SLIM task is an attempt at a concrete problem capturing “contextual recall”, where each number looks for the previous number of the same “type” and sums them.
>
> **Choice of Auxiliary Functions:** This is a great question. While the choice of auxiliary tasks currently *does* depend on the specific problem, we believe this is natural for problems with precise answers. Indeed, such problems require particular tasks to be done correctly and repeatedly, and hence it makes sense to robustly train for them. If we want to consider multiple problems simultaneously, one can go beyond the naive approach of training on the auxiliary tasks for each of these problems by use a Boosting/Hedge-based approach, where we adaptively train the model on a small set of auxiliary tasks on which the model has the worst performance.
>
> It is a very interesting question about how to isolate these auxiliary tasks automatically: one way would be to have a “library” of “basis” functions for the “local” tasks, and then use these as subroutines in a general training setup where they can hopefully enable length generalization for a variety of downstream applications.  Thank you for raising this point; we will add it to the discussion part of the revision.
>
> **Comparisons to Other Approaches:** In the revision, we will add the comparisons to the mentioned approaches.  Please note that we did not add those because the approaches are complementary to ours: curriculum training increases the complexity of the training data, whereas architectural constraints try to enforce additional structure to achieve their results. In our framework, our focus is quite different.
>
> **Extensions to “Fuzzy” Problems:** This is a great question. Our focus has been on “algorithmic” problems for which the auxiliary tasks can be naturally identified, and where length generalization can be precisely and objectively measured. Language modeling is a more amorphous task, and we feel that your question will require further research. E.g., one can imagine that for languages for which we do not have huge data corpuses, we should be able to use its grammar to define auxiliary tasks, and train the models to be better at handling intricate language constructs. (Our SLiM task tries to capture the kind of contextual recall typical in natural language.)
>
> **Further Experimental Comparisons:** We currently do not have experimental comparisons to curriculum training, since we consider it orthogonal to our focus. We will mention this in the “Limitations” section.

---

> > ### Comment · Reviewer_hBbS · 2025-08-07
> >
> > Thank you for the clarification. Yes, I was referring to arXiv:2310.00726, which shares significant overlap in terms of the core idea. One open issue that remains insufficiently addressed is the lack of a systematic approach for selecting "useful" auxiliary tasks. Regarding comparisons to other approaches, I acknowledge the point that the alternatives I previously mentioned are better viewed as complementary methods rather than direct baselines.

---

### Official Review · Reviewer_itBv · 2025-07-05

**Clarity:** 3
**Significance:** 3
**Originality:** 3
**Rating:** 4
**Confidence:** 3

**Summary:**

The paper introduces a new methodology for training transformers for length generalization, based on identifying auxiliary tasks which should improve the length generalization on the main task when trained on jointly. The proposed method improves the length generalization  performance on s-sparse tasks such as sorting, reversal, increment, SLiM, copy and parity as shown by a significant margin. This method is inspired by the theoretical analysis provided by the authors, which shows that under certain assumptions, the generalization error decays linearly in N.

**Questions:**

1) Could the authors make the connection between the theoretical analysis and the proposed training procedure clearer?

My current understanding is that for s-sparse functions - which are functions that are basically determined by s positions in the input sequence - you can introduce a notion of $(N, n)$ perturbations. This is the distribution over sequences of length $n+s$, obtained by sampling sequences of length $N$, fixing the s positions of the s-sparse function and then sampling the rest. And now you have a notion of robustness connecting the probability of mistakes on $D_n$ and the $(N, n)$ perturbations of $D_N$.

How does this analysis imply directly the training methodology you suggest?

2) Is there any connection between the proposed method and curriculum learning, for example the lenght curriculum proposed in [1]?

3) In Figure 5, is the amount of data fixed for the 2 tasks (main and auxiliary)? Could the improvement come from the fact that the AuxTask model has been trained on more unique data?

4) The choice of the auxiliary tasks seems too task-dependent: is there a general fix? How to adapt this strategy to general training on a mixture of tasks? How much is the gain sensitive to the choice of the auxiliary task?

5) Could the authors explain what is the auxiliary task for SLiM? It is not clear from Figure 4.

[1] Pouransari, Hadi, et al. "Dataset decomposition: Faster llm training with variable sequence length curriculum." Advances in Neural Information Processing Systems 37 (2024): 36121-36147.

**Ethical Concerns:**

["NO or VERY MINOR ethics concerns only"]

**Limitations:**

yes

**Quality:**

3

**Strengths And Weaknesses:**

Strengths:
- The theoretical analysis provided by the authors is sound and is able to predict the empirically observed results in Figure 6
- Across all synthetic tasks that the authors evaluate on, they obtain significant gains in performance as shown in Figure 5

Weaknesses:
- The main weakness of the paper is the limited empirical evaluation. The method proposed by the authors would be much stronger if it also worked in experiments on addition and multiplication, as well as experiments in more realistic architectures such as deeper networks (with MLPs) with an ablation over positional embeddings
- It is not clear to me how this method could be extended to other tasks, such as ListOps, or addition. Could the authors give examples of what would be an auxiliary task in these cases?

---

> ### Author Rebuttal · Authors · 2025-07-30
>
> We thank the reviewer for their encouragement and comments. We are glad that you liked the general framework, the fact that the experiments achieve the significant gains as suggested  by the theoretical model. Please find responses to your specific questions and comments below.
>
> **Clarification of the Setup**: Your statement regarding the problem setup is correct: we focus on sparse functions, consider $(N,n)$-perturbations (which are sequences of length $n$ produced by sub-sampling $N$-length sequences), and train for both the auxiliary and main tasks on these sequences. To complete the picture, the steps are as follows:
>
>    a. **Identifying Auxiliary Tasks**: For a sparse function, we produce multi-token outputs, where each token at timestep $t$ is obtained by applying some “local” function $f_t$ to a small number of previous tokens. Given the sparsity, there can only be a _bounded_ number of such functions; moreover, for many common tasks, there are just a handful of functions being repeatedly applied, e.g., finding the minimum and successor in sorting, or doing single-digit addition in the increment task, or summing two tokens in the SLiM task, etc. These commonly-used functions for a particular problem are natural auxiliary tasks for that problem.
>
>
>    b. **Robust Training for Auxiliary Tasks** : Having identified these auxiliary tasks, we train the model to be $(N,n)$-robust on auxiliary tasks (in addition to training on the main task). Recall that this training is on $n$-length sequences.
>
>
>    c. **Robustness implies Length Generalization**: We can now use our theoretical guarantees, which say that if we are $(N,n)$-robust with good parameters, then length generalization happens (i.e., low error on $N$-length sequences).
>
>
>  **Relationship to other Approaches:** We feel the ideas in our approach and in curriculum training are orthogonal. The latter focuses on training on progressively more complex instances (e.g., the approach of [1] buckets the sequences by their lengths, and then trains on some carefully chosen mixture of these sequences). Our approach is to identify useful building blocks (i.e., the auxiliary tasks) that can improve the training performance, and then to perform robust training for them (in addition to the main task). We feel the two approaches could be fruitfully combined in the future.  We will add a remark to this effect in the revision.
>
>
> **Data Usage Identical:** Yes, the amount of data is the same for the two approaches. So the baseline model is in total using the same amount of data as the proposed model (with data for main task + data for auxiliary task. On the point of unique data, it is an interesting comment: in fact, that is to some extent what our theory proposes—if we show more robust trajectories (which will be unique) to the model then it will length-generalize.
>
>
> **Choice of Auxiliary Functions:** This is a great question. While the choice of auxiliary tasks currently *does* depend on the specific problem, we believe this is natural for problems with precise answers. Indeed, such problems require particular tasks to be done correctly and repeatedly, and hence it makes sense to robustly train for them. If we want to consider multiple problems simultaneously, we can definitely go beyond the naive approach of training on the auxiliary tasks for each of these problems—we can use a Boosting/Hedge-based approach, where we adaptively train the model on a small set of auxiliary tasks on which the model has the worst performance.
>
> It is a very interesting question about how to isolate these auxiliary tasks automatically: one way would be to have a “library” of “basis” functions for the “local” tasks, and then use these as subroutines in a general training setup where they can hopefully enable length generalization for a variety of downstream applications. Thank you for raising this point; we will add it to the discussion part of the revision.
>
> **Clarification of Auxiliary Task for SLiM:** For SLiM the auxiliary task is as follows: given a valid sequence $S$ of tokens (produced as in the original problem), and a specific “query” token $q$, the auxiliary task is to lookup the last token $r$ in the input sequence $S$ of the same type as $q$, and to output $(q+r)$ modulo $100$. Note that this **contextual lookup/recall** is precisely the task the SLiM has to repeatedly perform using the last output token as its query, which is why it was chosen as an auxiliary task. Moreover, since contextual lookup/recall is an essential part of language, we think of this as a first yet concrete step towards understanding length generalization in broader contexts.

---

> > ### Comment · Reviewer_itBv · 2025-08-08
> > **Reply**
> >
> > Thank you very much for your replies! I will keep my grade.

---

### Note · Authors · 2025-08-15

We thank the reviewers. However, we contend that several critiques misinterpret the fundamental contribution and significance of our work.

Our paper is not merely another empirical method for length generalization. It provides the first formal theory explaining why decomposing tasks into sparse sub-computations is effective for length generalization. This theoretical framework is our core contribution and directly motivates a novel robust training recipe, a crucial distinction from prior intuition-based work like Awasthi and Gupta [AG23]. The superiority of our principled approach is not speculative; it is demonstrated by quantifiably better results (e.g., 1.5x vs 1.2x generalization on Increment).

Criticisms regarding our focus on synthetic tasks miss the point. This is a deliberate and necessary scientific choice. Controlled algorithmic tasks are the only way to isolate and objectively measure length generalization, a standard practice in this subfield.

---

### Decision · Program_Chairs · 2025-09-17

**Decision:**

Accept (poster)

**Comment:**

The paper provides novel theoretical framework proving length generalization results on a single layer Transformer. Specifically, the authors show that under an assumption of robustness to perturbation, any model with low error on "short" sequences will suffer moderately increasing error for longer sequences. Following this result, the authors show that this robustness assumption can be enforced during training by constructing an auxiliary task, such that when adding this auxiliary task to the training it improves length generalization capabilities on the original target task.

Reviewers appreciated the results and methods described in the paper, and overall found the theoretical framework novel and the underlying motivation to be important. One major criticism is the lack of proper reference to prior work, and specifically the work of Awasthi and Gupta [AG23], which empirically demonstrates the benefit of a very similar method of "task hinting" for length generalization. Additionally, the reviewers mentioned that there was no discussion on the effect of positional encoding and that the derivation of the auxiliary tasks is not clear. They also mention that there is some disconnect between the theoretical and empirical sections, and that the connection between them should be further clarified.

I believe that the theoretical framework and results introduced in this work are indeed valuable and do amount to significant contributions over prior results. However, the original manuscript has flaws as mentioned above, and in particular should have framed these nice theoretical contributions in the context of existing empirical works. I therefore recommend to accept the paper, assuming that the authors make the necessary changes to properly discuss prior works, as well as incorporating other feedback from the reviewers.